# Chlamydial protease-like activity factor targets SLC7A11 for degradation to induce ferroptosis and facilitate progeny releases

Wentao Chen[1,2‡], Xin Su[3‡], Yuying Pan[1,2‡], Han Zhou[1,2‡], Yidan Gao[4‡], Xuemei Wang[4], Lijuan Jiang[4], Lihong Zeng[1,2], Qingqing Xu[1,2], Xueying Yu[1,2], Xiaona Yin[1,2], Zhanqin Feng[1,2], Bao Zhang[5], Wei Zhao[5], Yaohua Xue[1,2*], Lingli Tang[4*], Heping Zheng[1,2,6*]

**1** Dermatology Hospital, Southern Medical University, Guangzhou, China, **2** Guangzhou Key Laboratory for Sexually Transmitted Diseases Control, Guangzhou, China, **3** Department of Clinical Laboratory, Guangdong Provincial Second Hospital of Traditional Chinese Medicine (Guangdong Provincial Engineering Technology Research Institute of Traditional Chinese Medicine), Guangzhou, China, **4** Department of Laboratory Medicine, The Second Xiangya Hospital of Central South University, Changsha, China, **5** Guangdong Provincial Key Laboratory of Tropical Disease Research, Key Laboratory of Infectious Diseases Research in South China of Ministry of Education, School of Public Health, Southern Medical University, Guangzhou, China, **6** Institute for Global Health, Southern Medical University, Guangzhou, China

‡ These authors share first authorship on this work.
* zhengheping@smu.edu.cn (HZ); linglitang@csu.edu.cn (LT); xueyaohua@smu.edu.cn (YX)

## Abstract

*Chlamydia trachomatis,* the most prevalent bacterial agent of sexually transmitted infections, poses a significant threat to reproductive health. The release of progeny through the orchestrated lysis of host cells plays a crucial role for the development of new infections, though the underlying molecular mechanisms remaining largely unexplored. In this study, we identified a novel mechanism by which *Chlamydia* induces host cell ferroptosis to facilitate its progeny release. This process involves the degradation of the host protein SLC7A11 by the chlamydial protease-like activity factor (CPAF), resulting in glutathione depletion and subsequent cell death characterized by lipid peroxidation. Infection with a CPAF-deficient strain fails to induce host cell ferroptosis. Notably, inhibiting ferroptosis by vitamin E reduces the *Chlamydia* burden in low genital tract of mice and trends toward attenuation of pathology. These findings provide new insights into the conserved survival strategies of *Chlamydia* and understanding of its pathogenesis.

## Author summary

*Chlamydia trachomatis* causes the most common bacterial sexually transmitted infection, trachoma, and lymphogranuloma venereum, and is linked to severe reproductive health issues such as pelvic inflammatory disease and infertility. This study uncovers a novel mechanism by which *Chlamydia* induces host ferroptosis. The chlamydial protease-like activity factor (CPAF) degrades SLC7A11, leading to glutathione depletion and lipid peroxidation, which triggers ferroptosis. Inhibition of ferroptosis reduced progeny release, decreased *Chlamydia* burden in the lower genital tract, and a trend toward attenuated

**Data availability statement:** The authors confirm that all data underlying the findings are fully available without restriction. All relevant data are within the paper and its Supporting Information files.

**Funding:** This work was supported by the National Natural Science Foundation of China (No. 81974307 to HZ), Guangzhou Science and Technology Plan Project (No. 202201000007 to HZ), Guangdong Basic and Applied Basic Research Foundation (No. 2021A1515012255 and 2023A1515012021 to WC; 2019A1515011827 to HZ; 2024A1515220022 to YX). This work was also partly supported by the Construction of the Pathogenic Microorganism (Toxin) Strain Resource Bank Capacity Project (LYF20230159 to LT) of The Second Xiangya Hospital, Central South University. The funders had no role in study design, data collection and analysis, decision to publish, or preparation of the manuscript.

**Competing interests:** The authors have declared that no competing interests exist.

pathology. These findings offer new insights into *Chlamydia's* survival strategies and pathogenesis.

## Introduction

*Chlamydia trachomatis* (CT) is responsible for the most common bacterial sexually transmitted diseases, as well as lymphogranuloma venereum and trachoma. *Chlamydia* infection is associated with severe reproductive health outcomes, including pelvic inflammatory disease (PID), ectopic pregnancy, infertility in women [1,2]. As an obligate intracellular bacterium, *Chlamydia* infection relies on the release of progeny from an infected cell to enter and infect a new host cell [3,4].

The progeny release of CT consists of at least two processes: extrusion and lysis [4–6]. The former is a conserved process involving actin polymerization and phosphorylated myosin light chain 2, myosin light chain kinase, myosin IIA, myosin IIB, and septins [7–9]. CT promotes or inhibits host cell death in specific contexts [10]; for example, CT prevents host cell death in order to evade immune surveillance [11]. However, the detailed mechanism of lysis at the late stage of chlamydial developmental cycle remains not fully explored.

Ferroptosis [12], a newly characterized form of programmed necrotic cell death resulting from impaired lipid peroxide repair systems, exerts significant influence on diverse physiological processes and various disease conditions, including infectious diseases [13–18]. Although the involvement of ferroptosis in the pathogenesis of CT has not been demonstrated so far, the replication of *Chlamydia* leads to the generation of reactive oxygen species (ROS), which subsequently causes membrane lipid peroxidation [19]. Moreover, supplementing *Chlamydia*-infected lambs with vitamin E, a potent inhibitor of ferroptosis [20–22], results in improved treatment outcomes [23]. The potential interplay between CT and host cells in the context of ferroptosis remains an area requiring further investigation, as suggestive evidence exists, but the specific mechanisms and the role of ferroptosis in *Chlamydia* infection have not yet been fully elucidated.

Chlamydial protease-like activity factor (CPAF) is a secreted protease produced by CT and other *Chlamydia* species. Functioning as a bacterial virulence factor, CPAF exhibits the ability to proteolytic degradation of proteins, including both host cell and chlamydial substrates [24–27]. This proteolytic activity allows CPAF to undermine host defense mechanisms, facilitate chlamydial survival, and evade host immune surveillance [24,27–29]. While CPAF has been implicated in the cell lysis process [6,29], the specific mechanism by which it operates remains poorly understood. In this study, we uncovered the critical role of CPAF in progeny release by triggering host cell ferroptosis through the degradation of SLC7A11. This process led to glutathione depletion, accumulation of lipid peroxidation, and subsequent host cell lysis. Pharmacological inhibition of ferroptosis reduced progeny release *in vitro*, decreased *Chlamydia* burden in low genital tract of mice, and showed a trend toward attenuation of pathology.

## Materials and methods

### Ethics Statement

All animal experiments in this study were approved by the Animal Welfare Committee of South China Agricultural University (2022G007) and conducted according to the guide for the care and use of laboratory animals.

## *Chlamydia* infection

HeLa-229 (CCTCC Cat# GDC0335) and McCoy cells (ATCC Cat# CRL-1696) were cultured in high-glucose Dulbecco's Modified Eagle's Medium (DMEM) (Thermo Fisher Scientific, Grand Island, NY, USA; 11995065) supplemented with 10% fetal bovine serum (Gibco; 10099141C) at 37°C and 5% $CO_2$. Prior to experimentation, the exclusion of *Mycoplasma* contamination was confirmed using EZ-PCR Mycoplasma Detection Kit (Biological Industries, Beit-Haemek, Israel; 20-700-20). To establish the infected cell model, cells were infected with CT serovars A, D, L1, L2, and *Chlamydia muridarum* (CM) at specific multiplicities of infection (MOIs).

## Pharmacological treatment

To inhibit the ferroptosis of host cells induced by *Chlamydia*, we added either ferrostatin-1 (10 µM; MedChemExpress, Shanghai, China; HY-100579), liproxstatin-1 (1 µM; Cayman Chemical, Ann Arbor, MI, USA; 17730–5), or trolox (3.2 mM; MedChemExpress, Shanghai, China; HY-101445) to the culture medium at the start of infection. To inhibit the potential proteolytic degradation during the late stage of *Chlamydia* infection, infected cells were treated with lactacystin (10 µM; Adipogen Life Sciences; San Diego, CA, USA; AG-CN2–0442-C100) or MG132 (10 µM; MedChem-Express; Shanghai, China; HY-13259) for 6 hours prior to the protein determination at 72 hours post-infection (h.p.i.). Cycloheximide (1 µg/mL; MedChemExpress, Shanghai, China; HY-12320) was added to the culture to accelerate cell death induced by *Chlamydia* during the ferrostatin-1, liproxstatin-1, and trolox treatment experiment. Three independent experiments were performed for analysis.

## Cell death assay

Necrotic cell death was evaluated by staining cells with 500 nM of propidium iodide (PI; Thermo Fisher Scientific, Waltham, MA, USA; P3566). Following a 5-minute staining period, PI-positive cells were visualized using inverted microscopy (Nikon, Tokyo, Japan; ECLIPSE Ti2). Cell viability was evaluated by measuring lactate dehydrogenase (LDH) release. The cell culture supernatant was collected and subjected to analysis using the CytoTox 96 assay (Promega, Madison, WI, USA; G1780) according to the manufacturer's instructions. Positive control wells were included to determine the maximum LDH release and calculate the percentage of LDH release. Three independent experiments were performed for analysis.

## Analysis of lipid ROS

Lipid peroxide production was assessed using the lipophilic fluorescent dye C11-BODIPY 581/591 (Thermo Fisher Scientific, Waltham, MA, USA; D3861). The cell culture supernatant was replaced with 1 mL of fresh DMEM containing 5 µL of BODIPY, and incubated for 30 minutes at 37°C. The cells were then harvested using TrypLE Express Enzyme (Thermo Fisher Scientific; Waltham, MA, USA; 12604021) after three washes with phosphate-buffered saline (PBS). Subsequently, the cells were resuspended in fresh PBS for flow cytometry analysis using a BD FACSCelesta cell analyzer (BD Biosciences, Franklin Lakes, NJ, USA). The oxidation of the polyunsaturated butadienyl portion of the dye induces a shift in the fluorescence emission peak from 590 nm to 510 nm. Mean fluorescence intensity (MFI) values were quantified for further statistical analysis. Three independent experiments were performed for analysis.

## Immunoblot analysis

Cultured cells or the culture supernatant was lysed using radioimmunoprecipitation assay (RIPA) buffer (Thermo Fisher Scientific, Waltham, MA, USA; 89900) supplemented with a protease and phosphatase inhibitor cocktail (Invitrogen; Thermo Fisher Scientific, Waltham, MA, USA; 78440). Subsequently, the samples were incubated with reducing sodium dodecyl sulfate-polyacrylamide electrophoresis loading buffer (CWBio, Beijing, China; CW0027) at 100°C for 10 minutes. To confirm that CPAF-mediated degradation of SLC7A11 is not influenced by post-lysis effects, we employed hot SDS buffer to lyse the cells, a method recommended in previous studies to eliminate such artificial effects [30]. Vimentin, a known CPAF substrate degraded due to post-lysis effects, was used as a control [26]. Antibodies against specific proteins were used as follows: GPx4 (1:1000, Abcam; ab125066), SLC7A11 (1:1000, Abcam; ab175186), major outer membrane protein (MOMP; 1:1000, Abcam; ab20881), and glyceraldehyde 3-phosphate dehydrogenase (GAPDH; 1:5000, Abcam; ab181602); fibroblast-specific protein 1 (FSP1; 1:500, Santa Cruz Biotechnology; sc-377120), TfR1(1:500, Santa Cruz Biotechnology; sc-32272), Vimentin (1:1000, Santa Cruz Biotechnology; sc-6260); horseradish peroxidase-conjugated anti-rabbit IgG (1:5000, Cell Signaling Technology; 7074S) and anti-mouse IgG (1:5000, Cell Signaling Technology; 7076S); mouse monoclonal antibody clones 100a against a 35 kDa activated CPAFc (kindly provided by Prof. Guangming Zhong, 1:50 dilution). Blots were imaged using a ChemiDoc imaging system (Bio-Rad; Hercules, CA, USA; v2.4.0.03). Three independent experiments were performed for analysis.

## Assessment of intracellular GSH

Intracellular glutathione (GSH) levels were measured using the glutathione kit (Abcam; Cambridge, MA, USA; ab112132) according to the manufacturer's instructions. Briefly, HeLa-229 cells were collected and washed with PBS, followed by incubation with the Green Dye (Component A) at 37°C for 30 minutes. The fluorescence intensity of the Green Dye was detected using flow cytometer at the FITC channel. High FITC fluorescence represents a high level of intracellular GSH, and the fraction of cells with high fluorescence was calculated. Three independent experiments were performed for analysis.

## Quantitative real-time PCR of *Chlamydia*

CT DNA was extracted from the culture supernatant or cell lysates using the TIANamp Genomic DNA kit (Tiangen Biotech, Beijing, China; DP304). All primers and probes used in this study were provided in **S1 Table**. Quantitative PCR assay was performed using TaqMan Gene Expression Master (Thermo Fisher Scientific, Waltham, MA, USA; 4369016) and the reaction conditions were as described in previous study [31]. The standard, primers, and probes were synthesized by Sangon Biotech (Shanghai, China).

## Evaluation of the relative mRNA expression of SLC7A11 and GPx4

Total RNA was extracted from *Chlamydia*-infected HeLa-229 cells at 72 hours post-infection using the TRIzol reagent (Thermo Fisher Scientific, Waltham, MA, USA; 15596018). First-strand cDNA was prepared using the Prime Script RT Reagent Kit (Takara Bio, Shiga, Japan; RR047B) according to the manufacturer's instructions. The relative mRNA expression was measured by TB Green Premix Ex Taq II (Takara Bio, Shiga, Japan; RR82WR) according to the manufacturer's instructions. Primer sequences were provided in **S1 Table**.

## Inducible CPAF expression in HeLa-229 cells

The coding sequence of CT CPAF (amino acid residues 18–601) was cloned into the pLVX-TetOne-Puro plasmid (P1686) obtained from MIAOLING Biology, Wuhan, China. The pLVX-TetOne-CPAF-puro plasmid, along with packaging plasmids, was co-transfected into HEK293T cells (CCTCC; GDC0187) to generate infectious lentivirus particles. The lentivirus particles were harvested from the cell culture supernatant. HeLa-229 cells were then transduced with the lentivirus in the presence of polybrene to facilitate efficient viral entry. To establish inducible CPAF expression in HeLa-229 cells, puromycin was added to the cell culture media to selectively pressure and generate stable cell lines. Similarly, transduction with the empty pLVX-TetOne-puro vector (vehicle) was performed as a control. The vehicle and CPAF-inducible expressed HeLa-229 cells were cultured overnight. CPAF expression was induced by treating the cells with 1 µg/mL doxycycline for 24 hours in preparation for subsequent experiments.

## Cell-free degradation assay

To measure CPAF degradation of SLC7A11, a cell-free degradation assay was performed. Fusion proteins were generated for the assay by cloning chlamydial DNA sequences encoding CPAF into a pET30a vector (constructed by Genscript Biotech), resulting in fusion proteins with a His-tag as the partner. HeLa-229 cells were lysed on ice for 10 minutes using RIPA buffer. The cleared lysates, which served as the source of host protein substrates, were obtained by centrifugation at 14,000×g for 15 minutes at 4°C. Recombinant wild-type CPAF or mutant CPAF (H105A) at various concentrations were pre-incubated with the HeLa-229 cell lysates at room temperature for 30 minutes, in the presence or absence of the proteasome inhibitor lactacystin (Adipogen Life Sciences; San Diego, CA, USA; AG-CN2–0442-C100) at the indicated concentration.

## Immunofluorescence assay

HeLa-229 cells grown on the 24 well plate with chlamydial infection were fixed and permeabilized for immunostainings. DAPI (Beyotime Biotech Inc; C1006) was used to visualize nuclear DNA. For titrating infection-forming units (IFU) from swab samples, a homemade mouse anti-*Chlamydia muridarum* polyclonal antibody plus a goat anti-mouse IgG H&L antibody (Alexa Fluor 555) (Abcam; ab150114) was used to visualize *Chlamydia* inclusions. For monitor the growth dynamic of CT, a mouse anti-chlamydial Hsp60 antibody conjugated to Alexa Fluor 488 (Santa Cruz; sc-57840 AF488) was used. All immunofluorescence-labeled samples were observed under confocal microscopy (Nikon, Tokyo, Japan; Model A1R).

## *Chlamydia muridarum* infected mouse model and monitoring

Female 6-week-old C57BL/6 mice were purchased and allowed to acclimate to the new facility for 1 week. One week prior to infection, the mice were gradually switched from standard chow to diets with different concentrations of vitamin E: low vitamin E (7 mg/kg) and high vitamin E (120 mg/kg), as previously used in ferroptosis inhibition animal study [20–22]. Since vitamin E is essential for normal mouse development, the low vitamin E diet was used as a control group to ensure proper growth [20]. Apart from the varying vitamin E content, the two diets were identical in all other components. Mice were maintained on these diets throughout the entire infection period. The infection model was established following a previously described protocol [32]. Five days prior to infection, mice were intramuscularly injected with 2.5 mg of depot medroxyprogesterone acetate (Abcam; ab142633). Mice were then inoculated with 2 ×

10$^5$ IFU of purified CM elementary bodies (EBs), diluted in 10 μL of SPG buffer, into the vaginal cavity. Lower genital tract infection was monitored through cervicovaginal swabbing and titration on the indicated days. Swabs were placed in 0.5 ml of sterile SPG medium containing sterile glass beads, vortexed for 2 min, and titrated via immunofluorescence assay as described above. Mice were anesthetized and euthanized by cervical dislocation at 56 days post-infection (d.p.i) for pathological examination.

### Pathology and inflammation assessments

Pathology and inflammation assessments were performed as described in previous studies [32]. Upon euthanasia, the mouse genital tracts were excised, and gross hydrosalpinx was evaluated using high-resolution digital photography. Hydrosalpinx severity was scored on an ordinal scale: 0-no hydrosalpinx, 1-observable only under magnification, 2-smaller than the oviduct, 3-equal to the oviduct, and 4-larger than the oviduct. Bilateral hydrosalpinx severity was calculated as the sum of the left and right oviduct scores, with incidence defined as the proportion of mice exhibiting a score ≥1. Chronic inflammation or mononuclear cell infiltration in the tissue surrounding the oviduct was assessed histologically. Genital tracts were fixed in 10% neutral formalin, stored in 70% ethanol, paraffin-embedded, and serially sectioned. Infiltration was scored on an ordinal scale: 0-no infiltration, 1-a single focus, 2-two to four foci, 3-more than four foci, and 4-confluent infiltration, following previously established guidelines [32]. The median score from three sections per oviduct was used to calculate the unilateral inflammation score, and the bilateral score was derived from the sum of both unilateral scores. The average diameter of the uterine horns was calculated from bilateral uterine horns in three sections.

### Statistical analysis

Data analysis and visualization were performed using Prism 8 software (GraphPad, La Jolla, CA, USA). Quantitative data were presented as mean ± standard deviation and were assessed for normality using the Shapiro-Wilk test. Statistical analyses are detailed in the respective figure legends. For multiple comparisons (more than two groups), a Bonferroni correction was applied. Statistical significance was determined at a threshold of $P<0.05$.

## Results

### *Chlamydia* triggers host cell ferroptosis in the late stage of developmental cycle

To investigate the lysis process during infection, we employed a model using HeLa-229 cells infected with CT serovar D (CT-D) at different multiplicities of infection (MOIs). We performed a time-course analysis to evaluate cell death by utilizing propidium iodide (PI) staining and lactate dehydrogenase (LDH) release assays. The results demonstrated that CT caused substantial cell death during the late stages of infection (Fig 1A-C). The temporal dynamics of *Chlamydia* growth are shown in **S1 Fig**, with a significant reduction in both the cell monolayer and *Chlamydia* inclusions at 72 h.p.i. Accumulation of lipid peroxides is a characteristic feature of ferroptosis [12]. To explore the induction of lipid peroxide accumulation by CT during the late stage of infection, we quantified the levels of lipid peroxidation in CT-infected cells using C11 BODIPY 581/591. Our findings revealed a significant time- and MOI-dependent increase in lipid peroxide levels within the infected cells (**Fig 1D** and **1E**). We also detected increased lactate dehydrogenase (LDH) release and production of lipid reactive oxygen species (ROS) in host cells infected with CT serovar L1 (CT-L1) and *Chlamydia muridarum* (CM), as illustrated in **Fig 1F** and **1G**.

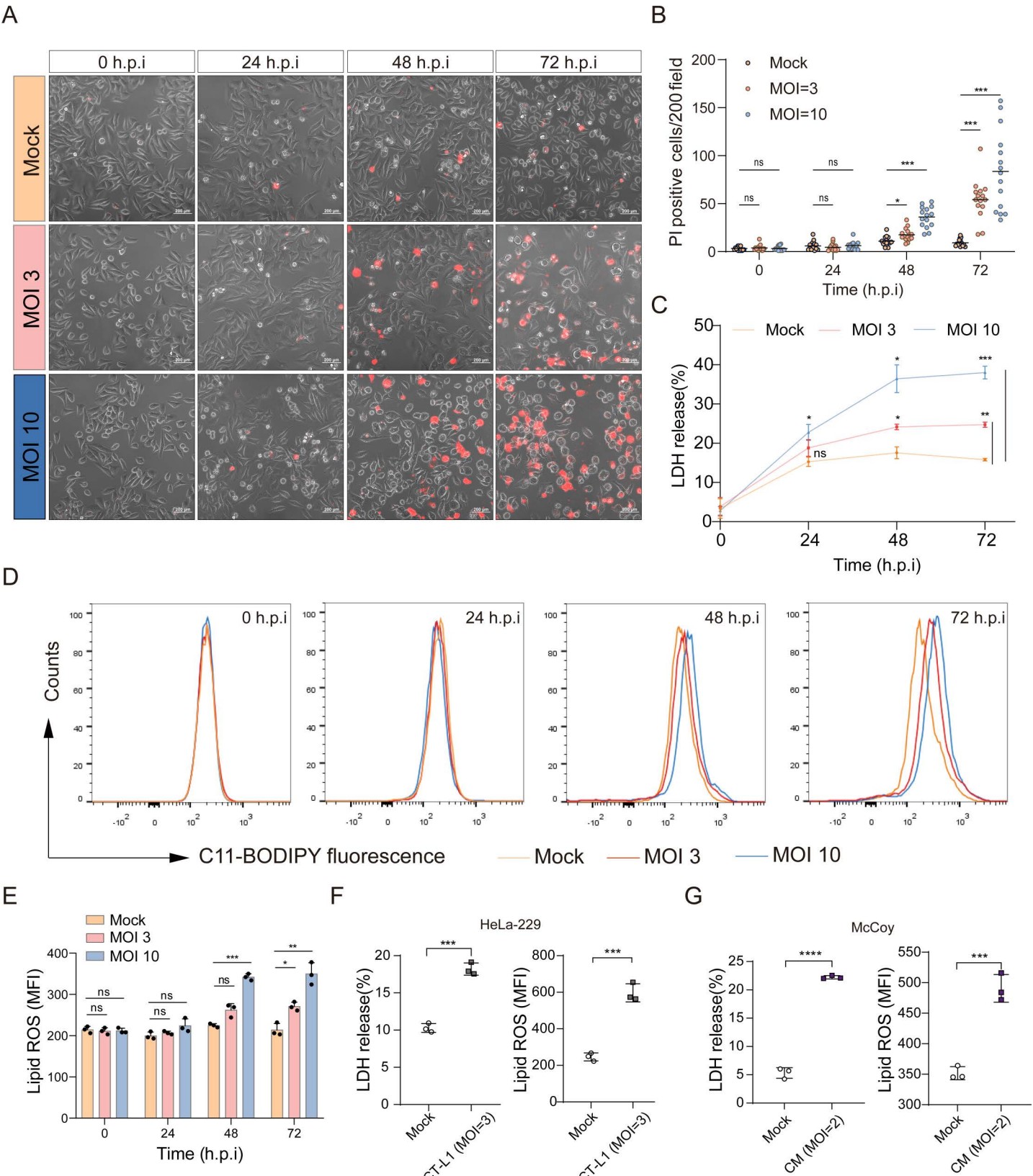

**Fig 1. *Chlamydia* triggers host cell ferroptosis during the late stage of its developmental cycle.** (**A**) Host cell death induced by *Chlamydia trachomatis* serovar D at various multiplicities of infection (MOIs) was assessed over a time course of 0–72 hours post-infection using a propidium iodide (PI) staining assay. Scale bars

represent 200 μm. (**B**) The number of PI-positive cells per 200× field was quantified for subsequent statistical analysis. Statistical significance was determined using a two-way ANOVA with Bonferroni's multiple comparisons. Individual data points represent the number of PI-positive cells per 200× field, with the line indicating the mean of positive cells across all 200× fields. Data were derived from three independent experiments, with five 200× fields analyzed per experiment (n=3). (**C**) The release of lactate dehydrogenase (LDH) was measured in both mock- and *Chlamydia trachomatis* serovar D-infected HeLa-229 cells over a time course of 0–72 hours post-infection. Statistical analysis was performed using a two-way ANOVA with Bonferroni's multiple comparisons. Individual data points represent the mean, and the bars represent the standard deviation (SD) of the mean (n=3). (**D**) The level of lipid peroxidation in *Chlamydia trachomatis* serovar D-infected cells at different MOIs throughout the infection was evaluated using C11-BODIPY staining, followed by flow cytometric analysis. (**E**) Statistical analysis of the data from (**D**) was conducted using a two-way ANOVA with Bonferroni's multiple comparisons. Data are presented as mean ± SD (n=3). (**F**) The release of LDH and the level of lipid ROS in *Chlamydia trachomatis* serovar L1 (CT-L1) (MOI 3)-infected HeLa-229 cells were measured and compared to mock-infected HeLa-229 cells (n=3). (**G**) The release of LDH and the level of lipid ROS in *Chlamydia muridarum* (CM) (MOI 2)-infected McCoy cells were measured and compared to mock-infected McCoy cells at 48 h.p.i. The Student's t-test was used for statistical analysis of (**F**) and (**G**). Data are presented as the mean ± SD (n=3). *, $P < 0.05$; **, $P < 0.01$; ***, $P < 0.001$; ns, not significant.

## Suppression of host cell ferroptosis impairs *Chlamydia* progeny release *in-vitro*

Ferrostatin-1, liproxstatin-1, vitamin E as well as its analogue trolox had been reported to prevent lipid peroxide-mediated ferroptosis (**Fig 2A**) [22,33]. We next examined whether the CT-driven cell death could be rescued by ferroptosis inhibitors. As shown in **Fig 2B**, treatment with ferrostatin-1 or liproxstatin-1 reduced LDH release and attenuated lipid peroxide accumulation, confirming that *Chlamydia* induces ferroptosis during its late developmental cycle. Lysis is the primary mechanism by which *Chlamydia* releases its progeny during the late developmental cycle [4]. We next investigated whether inhibiting ferroptosis could reduce the release of *Chlamydia* progeny. Immunoblot analysis of the major outer membrane protein (MOMP) revealed a significant reduction in chlamydial progeny levels in the cell supernatant following treatment with ferrostatin-1 and liproxstatin-1 (**Fig 2C**). This finding was further confirmed by quantitative qPCR targeting the cryptic plasmid of *Chlamydia* (**Fig 2D**). To exclude the possibility that the reduction in progeny levels in the supernatant was due to a direct inhibitory effect of the drugs on *Chlamydia* growth, we monitored the growth dynamics of *Chlamydia* under drug treatment compared to the control group. The results showed that neither ferrostatin-1 nor liproxstatin-1 significantly inhibited *Chlamydia* growth, except for ferrostatin-1, which demonstrated an inhibitory effect after 72 hours of treatment. (**Fig 2E**).

These effects were observed in both CT-L1- and CT-A-infected HeLa-229 cells with ferroptosis inhibition by liproxstatin-1, as shown in **Fig 2F** and **2G**. We also evaluated the ability of trolox, a water-soluble analogue of vitamin E, to inhibit *Chlamydia*-driven ferroptosis. Trolox demonstrated effects comparable to those of liproxstatin-1, significantly reducing LDH release, lipid ROS accumulation, and progeny release in the supernatant of CM-infected cells (**Fig 2**H-J). Additionally, the potential for trolox-induced inhibition of *Chlamydia* growth was ruled out, as confirmed in **Fig 2J**. These findings demonstrate the conserved nature of this mechanism across *Chlamydia* strains, where it hijacks host cell ferroptosis, a process that can be suppressed by ferroptosis inhibitors.

## Ferroptosis inhibitor vitamin E reduces *Chlamydia* burden and pathology in the mouse genital tract

Considering that inhibition of ferroptosis reduces progeny release, we hypothesized that ferroptosis inhibition would limit progeny release *in-situ*, thereby impairing the establishment of new infections in the surrounding area and ultimately leading to a suppressed infection *in vivo*. To investigate the role of ferroptosis in infection dynamics and the development of hydrosalpinx, C57BL/6 mice were intravaginally inoculated with CM and fed two different concentrations of vitamin E, which has been previously reported to inhibit ferroptosis [20–22]. By 56 d.p.i., mice on the high vitamin E diet exhibited a lower incidence of

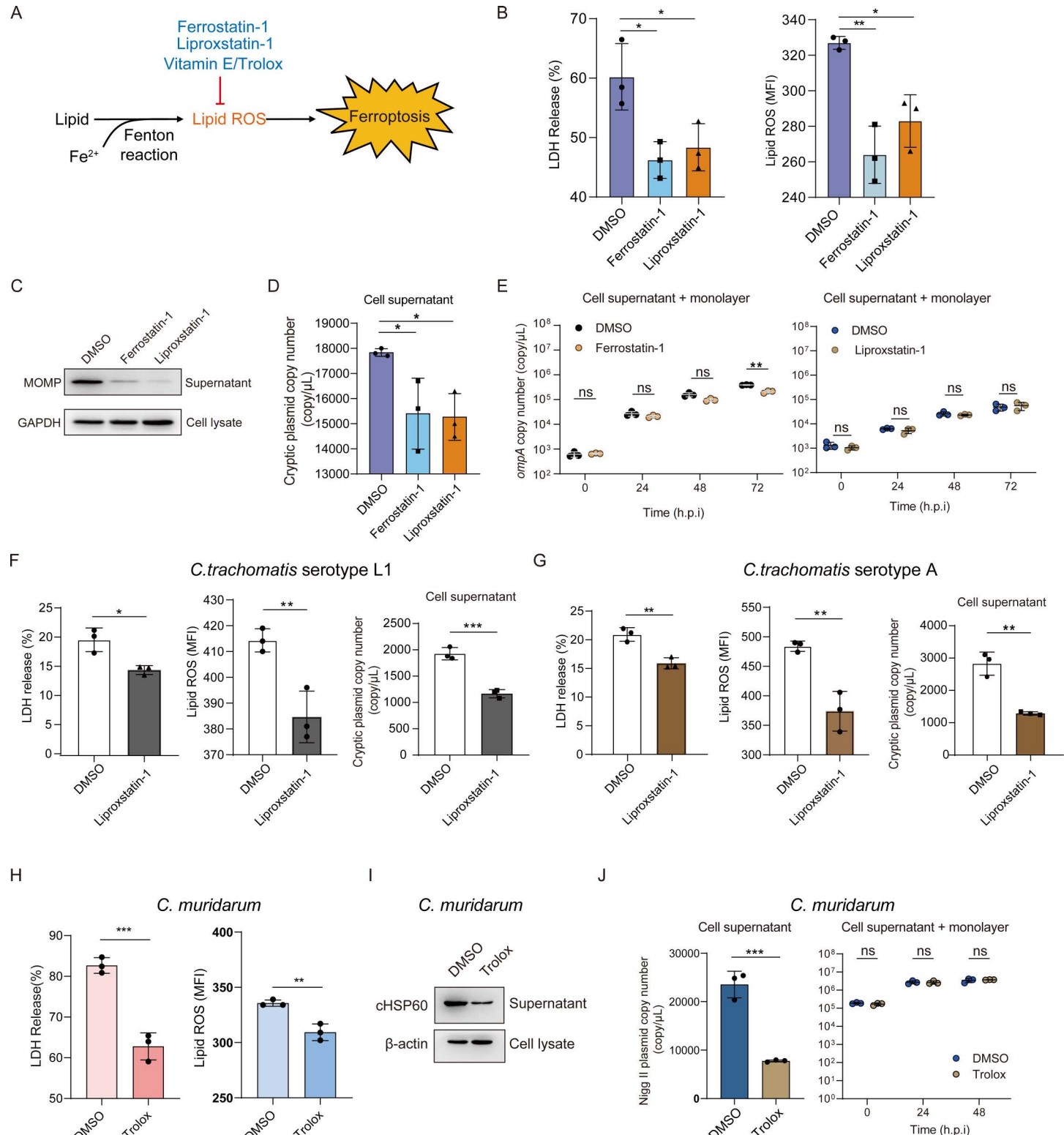

**Fig 2. Suppression of host cell ferroptosis impairs *Chlamydia* progeny release *in-vitro*.** (**A**) A schematic diagram illustrates a series of reported pharmaceuticals that block ferroptosis by inhibiting lipid ROS accumulation. (**B**) The release of LDH and the levels of lipid ROS in *Chlamydia trachomatis* serovar D (MOI 10)-infected HeLa-229 cells were assessed following treatment with ferrostatin-1 (10 μM) and liproxstatin-1 (1 μM) for 72 hours. Statistical analysis was conducted using a one-way ANOVA with Bonferroni's multiple comparisons (n=3). (**C**) Immunoblot analysis of chlamydial MOMP from cell supernatant and GAPDH from cell lysate were

conducted following treatment with ferrostatin-1 (10 μM) and liproxstatin-1 (1 μM) for 72 hours. (**D**) The copy number of the chlamydial cryptic plasmid in the cell supernatant of *Chlamydia trachomatis* serovar D (MOI 10)-infected cells was determined following treatment with ferrostatin-1 (10 μM) and liproxstatin-1 (1 μM) for 72 hours. Statistical analysis was conducted using a one-way ANOVA with Bonferroni's multiple comparisons (n=3). (**E**) The copy number of *ompA* in the total culture (cell supernatant and monolayer) of *Chlamydia trachomatis* serovar D (MOI 10)-infected cells was determined following treatment with ferrostatin-1 (10 μM) and liproxstatin-1 (1 μM) over a time course. Statistical analysis was conducted using a two-way ANOVA test (n=3). (**F, G**) The release of LDH, lipid ROS levels, and the copy number of the chlamydial cryptic plasmid of cell supernatant in *Chlamydia trachomatis* serovars L1 (MOI 3)- (**F**) and A (MOI 3)- (**G**) infected cells were measured following treatment with liproxstatin-1 (1 μM) for 72 hours. The Student's t-test was used for statistical analysis of (**F**) and (**G**) (n=3). (**H**) The release of LDH and the level of lipid ROS in *Chlamydia muridarum* (CM) (MOI 2)-infected McCoy cells were measured following treatment with trolox (3.2 mM) for 48 hours. The Student's t-test was used for statistical analysis (n=3). (**I**) Immunoblot analysis of chlamydial HSP60 in the cell supernatant and β-actin in the cell lysate was performed following treatment with trolox (3.2 mM) for 48 hours. (**J**) The copy number of the Nigg II plasmid in the cell supernatant and total culture (cell supernatant and monolayer) of CM (MOI 2)-infected McCoy cells was quantified over a time-course following treatment with trolox (3.2 mM). The Student's t-test was used for statistical analysis of the cell supernatant data (left panel), while a two-way ANOVA was applied to the data from the total culture (cell supernatant and monolayer) over the time course (right panel). Data are presented as the mean ± SD (n=3). *, $P < 0.05$; **, $P < 0.01$; ***, $P < 0.001$; ns, not significant.

hydrosalpinx compared to the control group on the low vitamin E diet (**Fig 3A** and **3B**). The severity of hydrosalpinx in mice on the high vitamin E diet was lower, although this difference was not statistically significant (**Fig 3C**). Cervicovaginal swabbing and titration during the infection revealed that mice fed the high vitamin E diet had a significantly reduced genital tract burden compared to the low vitamin E group, with the burden in the high vitamin E group being approximately half or even less than half of that observed in the low vitamin E group from 10 d.p.i. (**Fig 3D**). Additionally, mice in the high vitamin E group exhibited earlier pathogen clearance (**Fig 3D**). The level of inflammatory cell infiltration into the tissue surrounding the oviduct was also assessed, revealing that mice fed the high vitamin E diet had fewer inflammatory foci compared to the low vitamin E group, although this difference was not statistically significant (**Fig 3E** and **3F**). No significant difference in the diameter of the uterine horns was observed between the two groups (**Fig 3G**). Overall, inhibition of ferroptosis by vitamin E attenuated infection dynamics and showed a trend toward reduced development of hydrosalpinx.

## *Chlamydia trachomatis* triggers host cell ferroptosis through SLC7A11 downregulation and consequent glutathione depletion

In order to elucidate the mechanism by which *Chlamydia* manipulates ferroptosis, we assessed the involvement of two key cellular antioxidant systems, namely the System Xc−/glutathione (GSH)/GPx4 and FSP1/CoQ pathways [34–37], which play a critical role in defending against ferroptosis, during the late stage of CT infection. We observed a significant reduction in the abundances of SLC7A11 and GPx4 in cells infected with CT-D compared to mock-infected cells (**Fig 4A**). However, we did not observe a significant change in the levels of FSP1 under these conditions. The substantial decrease in the abundance of SLC7A11 and GPx4 were confirmed in CT-L1 and CM infected host cells (**Fig 4B**). SLC7A11 serves as a membrane antiporter, facilitating the uptake of cystine and enabling the subsequent synthesis of GSH, which collaborates with GPx4 in exerting anti-ferroptotic effects (**Fig 4C**) [35]. Therefore, we proceeded to investigate whether the level of intracellular GSH in infected cells was affected by the CT-induced deficiency of SLC7A11. As shown in **Fig 4D**, CT-infected cells displayed a substantial decrease in GSH levels compared to mock-infected cells. These findings support the involvement of the SLC7A11-GSH-GPx4 axis in *Chlamydia*-induced ferroptosis.

## CPAF mediates chlamydial-induced ferroptosis through proteolytic degradation of SLC7A11 in host cells

We observed the mRNA expressions of SLC7A11 and GPx4 were upregulated in infected cells (**Fig 5A**), which is consistent with the pattern of gene expression changes associated

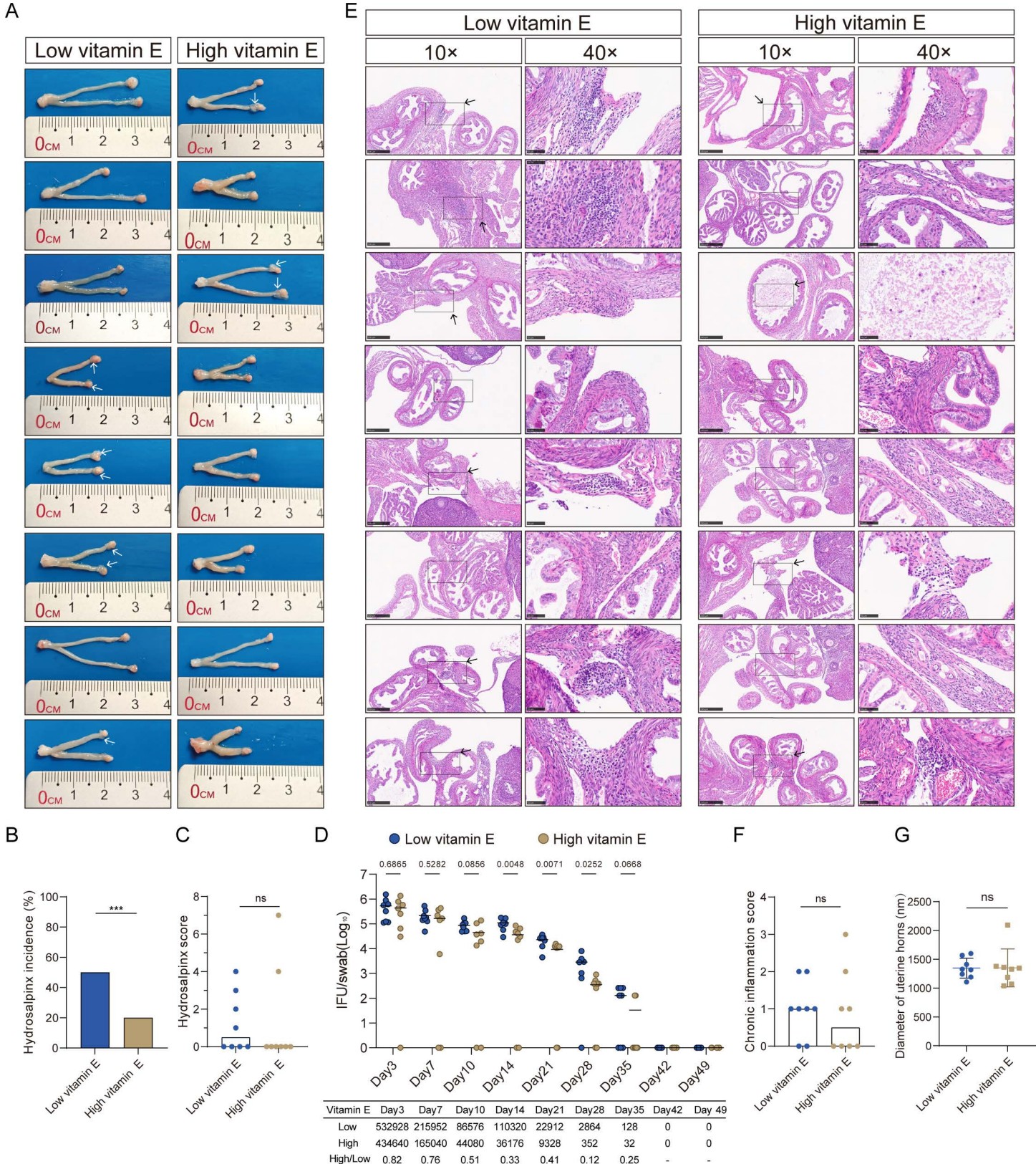

**Fig 3. Ferroptosis inhibitor vitamin E reduces *Chlamydia muridarum* burden and pathology in the mouse genital tract.** (A) Gross pathology images of whole genital tracts from *Chlamydia muridarum*-infected C57BL/6 mice, fed natural ingredient diets containing low (7 mg/kg) or high (120 mg/kg) vitamin E, were collected

at 56-day post-infection (d.p.i.). White arrow indicates hydrosalpinx (n = 8 per group) (**B**) Incidence of hydrosalpinx at 56 d.p.i. (***, *P* < 0.001, by Fisher's exact test; n = 8 per group). (**C**) Severity of hydrosalpinx at 56 d.p.i. Both individual bilateral scores (dots) and medians (bars) are shown. Statistical analysis was performed using the Mann-Whitney test (n = 8 per group). (**D**) Lower genital tract shedding course of *Chlamydia muridarum* following intravaginal inoculation with $2 \times 10^5$ IFU. Shedding was quantified from cervicovaginal swabs collected on the indicated days post-infection. Points represent the IFU of infectious EBs collected from each mouse, and the lines represent the means of 8 mice per group. Statistical analysis was performed using a two-way ANOVA. The *P* value is indicated in the panel. The table (bottom) shows the mean IFU and the ratio between the two groups on the indicated days (n = 8 per group). (**E**) Micrographs of H&E-stained oviducts from infected mice at 56 d.p.i. For each infected mouse, both a broad view of tissue sections (10×; scale bar, 250 μm) and an amplified view (40×; scale bar, 50 μm) are shown. The rectangular frame with an arrow indicates foci of chronic inflammatory cell infiltration (n = 8 per group). (**F**) Bilateral chronic inflammatory cell infiltrate scores were determined from H&E-stained tissue sections. Statistical analysis was performed using the Mann-Whitney test (n = 8 per group). ns, not significant. (**G**) Diameter of uterine horns was measured from H&E-stained tissue sections. Statistical analysis was performed using the Mann-Whitney test (n = 8 per group). ns, not significant.

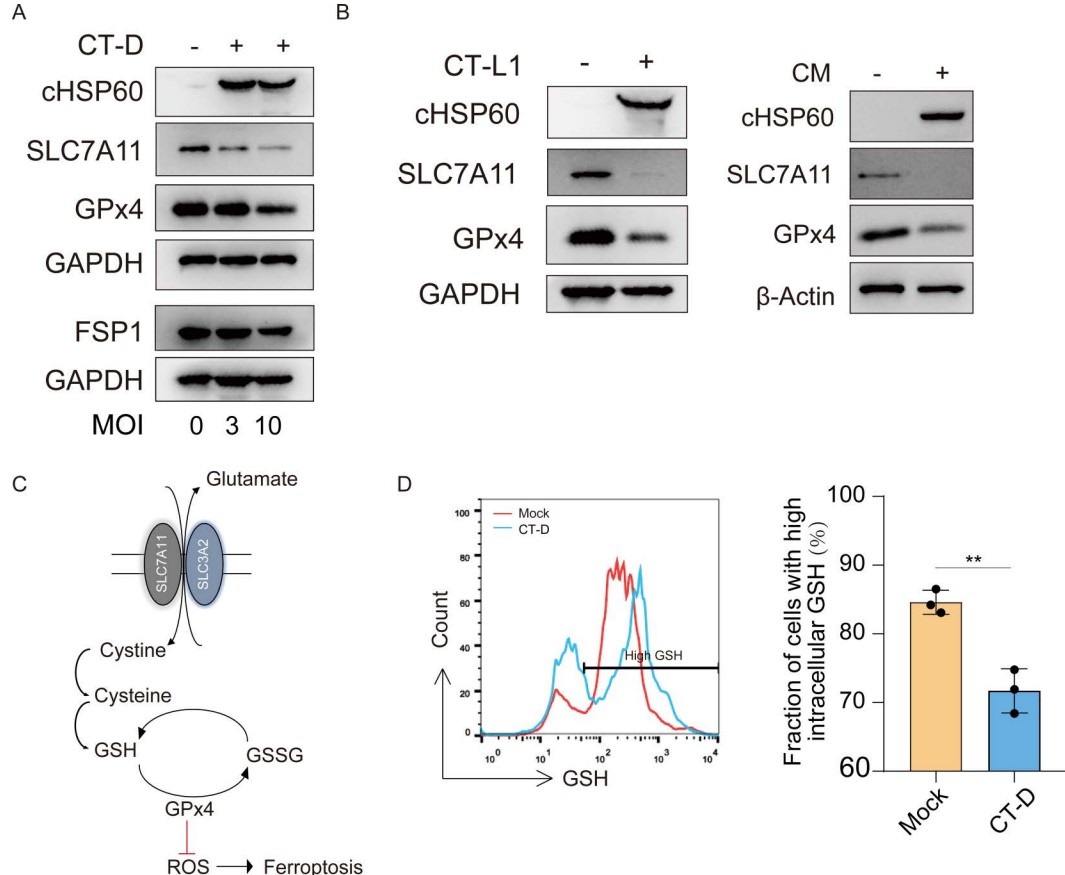

**Fig 4. *Chlamydia trachomatis* triggers host cell ferroptosis through SLC7A11 downregulation and consequent glutathione depletion.** (A) Immunoblot analysis of multiple ferroptosis-associated proteins and chlamydial HSP60 was performed in cells infected with various MOIs of *Chlamydia trachomatis* serovar D (CT-D) at 72 h.p.i., compared to mock-infected cells. (B) Immunoblot analysis of SLC7A11, GPx4, GAPDH, and chlamydial HSP60 was conducted in cells infected with *Chlamydia trachomatis* serovar L1 (CT-L1) (MOI 3) and *Chlamydia muridarum* (CM) (MOI 2), compared to mock-infected cells. (C) A schematic representation of the SLC7A11-GSH-GPx4 pathway in the regulation of ferroptosis. (D) Intracellular glutathione (GSH) levels were measured in CT-D (MOI 5)-infected cells at 72 h.p.i by flow cytometry, compared to mock-infected cells. The fraction of cells with high intracellular GSH was calculated. The Student's t-test was used for statistical analysis. Data are presented as the mean ± SD (n=3). **, *P* < 0.01.

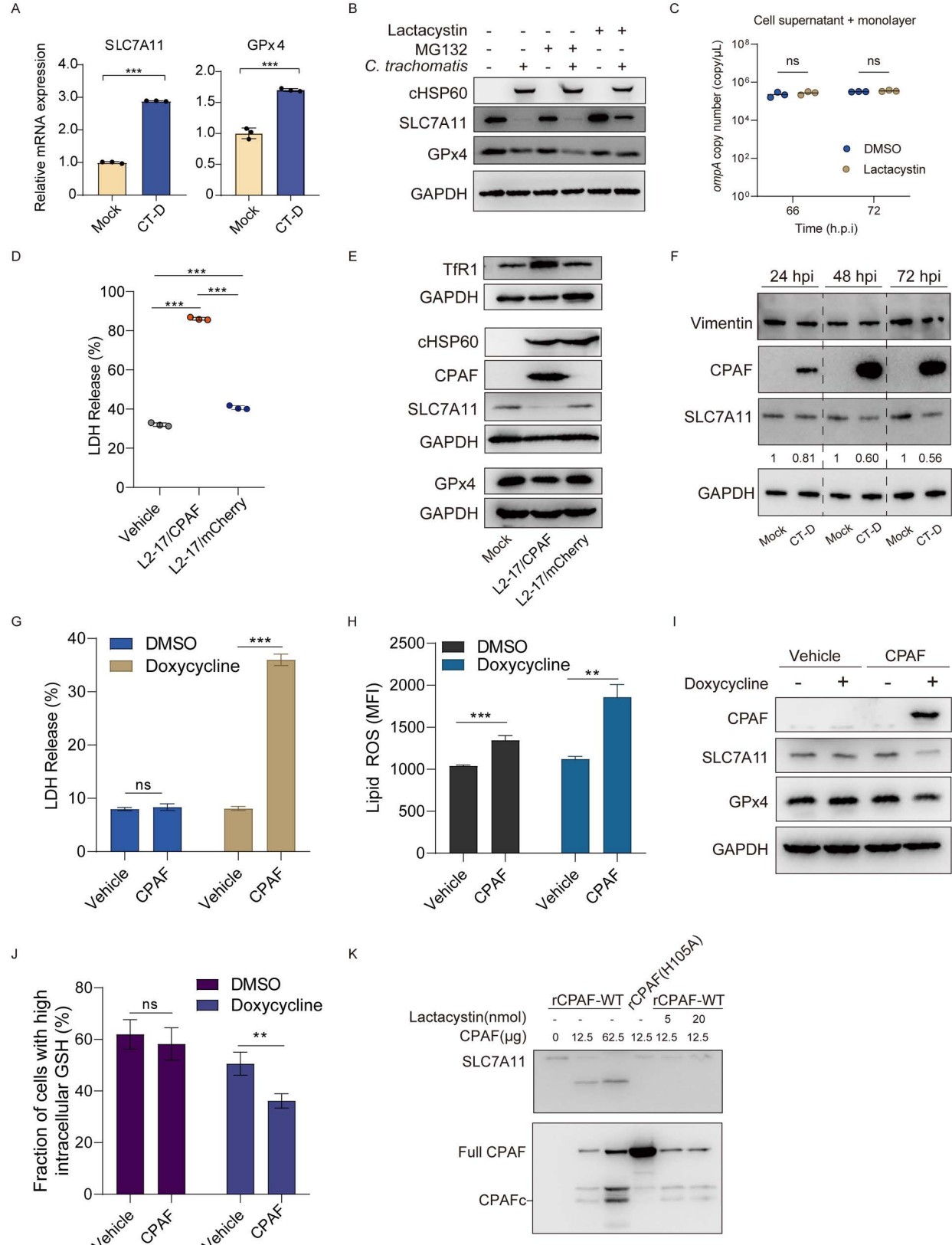

**Fig 5. CPAF mediates chlamydial-induced ferroptosis through proteolytic degradation of SLC7A11 in host cells.** (**A**) The mRNA expression of SLC7A11 and GPx4 in *Chlamydia trachomatis* serovar D (MOI 10)-infected cells was measured by SYBR Green qPCR at 72 h.p.i., compared to

mock-infected cells. Statistical analysis was performed using Student's t-test. Data are presented as the mean ± SD (n=3). (**B**) Immunoblot analysis of SLC7A11, GPx4, and cHSP60 in CT-D (MOI 10) infected cells was performed at 72 h.p.i., following a 6-hour pre-treatment with lactacystin (10 μM) or MG132 (10 μM). (**C**) The copy number of *ompA* in the total culture (cell supernatant and monolayer) of *Chlamydia trachomatis* serovar D (MOI 10)-infected cells was determined before and after treatment with lactacystin (10 μM). Statistical analysis was performed using a two-way ANOVA test (n=3). (**D**) The release of LDH from cells infected with the CPAF-deficient strain (L2-17/mCherry) (MOI 1) or the CPAF-supplemented strain (L2-17/CPAF) (MOI 1) was measured at 72 h.p.i. Statistical analysis was performed using a one-way ANOVA test with Bonferroni's multiple comparisons. Data are presented as the mean ± SD (n=3). (**E**) Immunoblot analysis of SLC7A11, TfR1, GPx4, GAPDH, and CPAF was performed across three groups: mock infection, CPAF-deficient strain (L2-17/mCherry) (MOI 1) infection, and CPAF-supplemented strain (L2-17/CPAF) (MOI 1) infection at 72 h.p.i. (**F**) Immunoblot analysis of SLC7A11, vimentin, CPAF, and GAPDH in *Chlamydia trachomatis* serovar D (MOI 3) -infected cells was performed using the hot SDS loading buffer lysis protocol over the course of infection. (**G**), (**H**) The release of LDH (**G**) and the level of lipid ROS (**H**) in HeLa-229 cells following the induction of CPAF expression by DMSO or doxycycline treatment were quantified. Statistical analysis was performed using Student's t-test. Data are presented as the mean ± SD (n=3). (**I**) The expression of CPAF and the degradation of SLC7A11 and GPx4 were confirmed in HeLa-229 cells after a 24-hour treatment with doxycycline, as determined by western blotting. (**J**) Intracellular glutathione (GSH) levels were determined in HeLa-229 cells following the induction of CPAF with doxycycline treatment. Statistical analysis was performed using Student's t-test. Data are presented as the mean ± SD (n=3). (**K**) Immunoblot analysis was performed to examine the degradation of SLC7A11 in cell lysates following a 30-minute incubation with recombinant wild-type CPAF (rCPAF) or the H105A mutant. An activated CPAFc fragment was observed in the wild-type CPAF group, but not in the CPAF (H105A) group. The degradation of recombinant wild-type CPAF was blocked by lactacystin. **, *P*< 0.01; ***, *P*<0.001; ns, not significant.

with ferroptosis [38]. This suggests that *Chlamydia* did not alter the transcriptional regulation of SLC7A11 and GPx4 during this process. Hybiske *et al* [4] previously described that the lysis activity of CT-infected cells was relative to proteases activity. To assess the potential involvement of protease activity in the reduced abundances of SLC7A11 and GPx4, we treated the infected cells with two proteasomal inhibitors, MG132 and lactacystin. Interestingly, only treatment with lactacystin resulted in the restoration of SLC7A11 and GPx4 levels (**Fig 5B**). We also excluded the potential inhibition of *Chlamydia* growth by lactacystin (**Fig 5C**). Previous studies have demonstrated the specific inhibitory effect of lactacystin, but not MG132, on the protease activity of chlamydial protease-like activity factor (CPAF) [39]. CPAF has been reported to have the capacity to induce necrotic cell death [40]. Based on these clues, we propose that CPAF may play a role in mediating this process.

To investigate whether CPAF contributes to driving ferroptosis in the late stages of CT infection, we used two transgenic strains, L2-17/mCherry (CPAF-deficient) and L2-17/CPAF (CPAF-complemented), to examine *Chlamydia*-driven ferroptotic events. Both transgenic organisms were generated by transforming the pGFP::SW2 plasmid, which expresses a wild-type allele of CPAF or mCherry, into the CPAF-deficient strain L2-17, as previously described [28,41]. A substantial decrease in LDH release was observed in cells infected with the CPAF-deficient strain L2-17/mCherry compared to the CPAF-complemented strain L2-17/CPAF (**Fig 5D**). The presence of the GFP coding element on the pGFP::SW2 plasmid in both transgenic organisms hindered the practical application of BODIPY 581/591 C11 for lipid peroxide detection. As an alternative, we employed TfR1 as a specific ferroptosis marker to investigate the potential role of CPAF in triggering ferroptosis [42]. The expression of TfR1 was significantly upregulated in cells infected with CPAF-expressing strains (L2-17/CPAF), whereas infection with the CPAF-deficient strain L2-17/mCherry showed similar expression levels to mock infection (**Fig 5E**). In line with this, the decreased levels of SLC7A11 and GPx4 were correlated with the presence of CPAF. The CPAF-deficient strain CT-17/mCherry did not exhibit alterations in the protein levels of these anti-ferroptosis factors (**Fig 5E**). Although the CPAF-deficient strain (CT-17/mCherry) exhibited impaired progeny release, as indicated by reduced levels of chlamydial HSP60 and cryptic plasmid in the supernatant (**S2A Fig**), it also showed attenuated growth kinetics compared to the CPAF-complemented strain (**S2B Fig**).

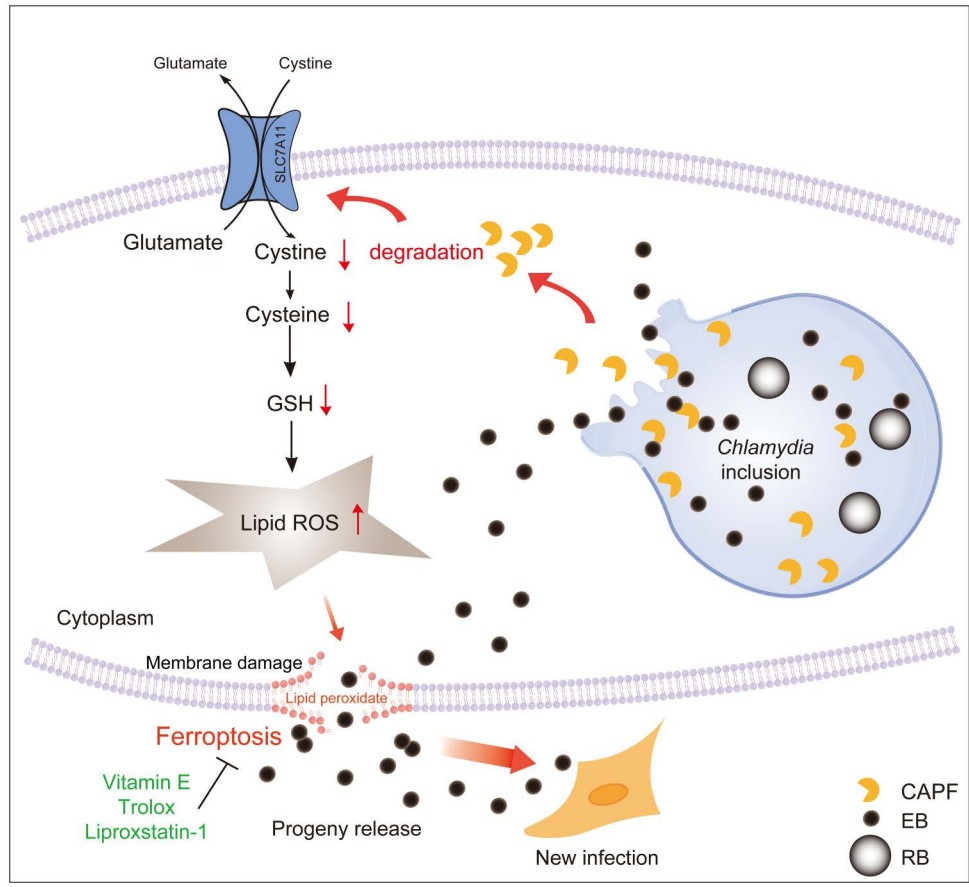

**Fig 6. Chlamydial protease-like activity factor targets SLC7A11 for degradation to induce ferroptosis and facilitate progeny releases.** A schematic diagram illustrates the key findings of this study, showing how Chlamydial protease-like activity factor directly degrades the host protein SLC7A11, leading to GSH depletion, lipid ROS accumulation of lipid ROS, and ultimately triggering host cell ferroptosis, which facilitates *Chlamydia* progeny release.

To confirm that CPAF-mediated degradation of SLC7A11 was not affected by post-lysis effects, we assessed SLC7A11 levels at multiple time points (24–72 h.p.i) using a protocol that blocks CPAF post-lysis effects [30]. As shown in **Fig 5F**, SLC7A11 degradation was consistently accompanied by an increase in CPAF levels. In contrast, vimentin identified as a substrate undergoing CPAF post-lysis degradation remained unchanged, serving as an experimental control [26].

To investigate whether CPAF alone could induce ferroptotic events, we developed a eukaryotic HeLa-229 cell line with a tetracycline (Tet) inducible expression system for CPAF. Following a 24-hour treatment with doxycycline, we observed increased LDH release and lipid ROS levels in the CPAF-expressed cells (**Fig 5G and 5H**), along with decreased abundances of SLC7A11 and GPx4 (**Fig 5I**). Additionally, the cellular GSH level, downstream of SLC7A11, was significantly inhibited in the CPAF-expressed cells compared to controls (**Fig 5J**). These findings suggest that CPAF alone is sufficient to induce ferroptotic events.

To investigate whether CPAF can directly cleave SLC7A11 and GPx4, we expressed recombinant CPAF (rCPAF) and a mutant form of CPAF (H105A) that resulted in a defective proteolytic activity [41,43]. To inhibit the proteolytic activity of rCPAF, lactacystin was employed. As shown in **Fig 5K**, rCPAF underwent autocatalytic cleavage, resulting in the generation of

activated CPAFc fragments. In contrast, the mutant rCPAF (H105A) did not produce the activated CPAFc fragment. SLC7A11 was degraded by incubating rCPAF but not rCPAF (H105A) or rCPAF combined with lactacystin treatment (**Fig 5K**). Notably, rCPAF didn't degrade GPx4 *in vitro* (**S3 Fig**).

## Discussion

While programmed cell death is widely acknowledged as a critical defense mechanism in the host's response to pathogen infection, pathogens have evolved sophisticated strategies to manipulate and subvert these host processes, turning them into advantageous mechanisms for their survival and propagation during the course of infection [44]. Lysis is a crucial mechanism for *Chlamydia* to release progeny and establish new infections [4], yet the underlying molecular mechanisms driving this process remain not fully unexplored. In this study, we unveiled a novel mechanism by which *Chlamydia* induces host cell ferroptosis to facilitate its progeny release (**Fig 6**).

The mechanism we uncovered represents a conserved strategy for *Chlamydia*, which has been observed across several serotypes and host cell types. Pathogens, particularly viruses, manipulate the lipid ROS process by targeting the GPX4-dependent antioxidant pathway, often through transcriptional regulation that disables the SLC7A11-GSH-GPX4 axis to promote propagation [13,17,18]. For example, *Hepatitis B virus* uses HBx to recruit EZH2 and H3K27me3 to the SLC7A11 promoter, suppressing its expression [18], while *Mycobacterium tuberculosis* employs PtpA to alter the GPX4 promoter via PRMT6-mediated modifications, reducing GPX4 expression [17]. In contrast, our study reveals that *Chlamydia* induces host cell ferroptosis by directly degrading SLC7A11, rather than through transcriptional regulation.

After controlling for potential *Chlamydia* growth inhibition effects, several ferroptosis inhibitors were found to reduce *Chlamydia* progeny release during the late stages of infection. *In vivo*, mice infected with CM and fed a high-vitamin E diet (a ferroptosis inhibitor) exhibited significantly lower bacterial burden in the lower genital tract and faster infection clearance compared to those on a low-vitamin E diet. Moreover, the incidence of hydrosalpinx was lower in the high-vitamin E group. Although no statistically significant differences were observed in the severity of hydrosalpinx or inflammatory infiltration, this may be attributed to the attenuated virulence of the CM strain used, resulting in generally milder pathology across all groups. Nonetheless, high-vitamin E-fed mice displayed a trend toward reduced pathology. Our study supported that the reports supplementing *Chlamydia*-infected lambs with vitamin E results in improved treatment outcomes [23].

CPAF, a serine chlamydial protease, harbors multiple functions in promoting CT survival by cleaving a broad of host and chlamydial substrates [24,27,29,45]. Despite previous proposals regarding CPAF's involvement in the release of CT progeny [6,29], the specific molecular mechanism underlying its role remains poorly understood. In the current study, we identified CPAF induces host cell ferroptosis through the direct proteolytic degradation of SLC7A11. Previous studies have identified inauthentic targets as CPAF substrates due to post-lysis protease effects; thus, we used hot SDS lysis to block these effects [26]. Our results confirmed that SLC7A11 was processed by CPAF within the cell, eliminating the potential confounding effects of post-lysis processes. SLC7A11 identified in this study also matches the dataset of CPAF substrates (not functionally validated) determined through a proteomic approach in previous studies [27]. From a functional validation perspective, the observation that depletion of SLC7A11 resulted in a significant reduction in intracellular GSH levels, which is imported into the cell via the xCT transporter formed by SLC7A11 and SLC3A2 in both CT-infected and CPAF-expressed cells, further supports the idea that the degradation of SLC7A11 was not caused by post-lysis protease effects mediated by CPAF. Instead, it suggests that the

degradation of SLC7A11 is an integral part of the infection process itself. Our results also demonstrate that GPx4 was suppressed in CT-infected or CPAF-expressed cells, but it was not degraded by recombinant CPAF in vitro, indicating that GPx4 is not a substrate of CPAF. A possible explanation for the reduced abundance of GPx4 in infected or CPAF-expressed cells is that the degradation of SLC7A11 by CPAF limits selenium uptake and selenocysteine synthesis, ultimately leading to the inhibition of selenoprotein GPx4 synthesis [46].

The deficiency of CPAF impaired chlamydial survival in the mouse lower genital tract [28]. Previous study reported that CPAF acted as an effector that paralyzes polymorphonuclear neutrophils upon *Chlamydia* exposure to host immunosurveillance, evading the host's innate immune response by suppressing the oxidative burst, interfering with neutrophil activation, and cleaving FPR2 [24]. In our study, we observed that the absence of CPAF in *Chlamydia* prevents the induction of ferroptosis in host cells, which impairs progeny release, may lead to a lower bacterial burden and faster clearance *in vivo*. However, due to the complexity of the context, where CPAF-deficient strains exhibit weaker growth kinetics compared to CPAF-complemented strains across multiple studies [28,47]. It is difficult to determine whether the inability to induce ferroptosis limits progeny release, or if the reduction in progeny production is due to CPAF deficiency, or whether a weakened resistance to immune clearance is at play, or a combination of these factors. Our study was limited to investigating the relationship between CPAF and ferroptosis *in vitro*.

Our study did not prioritize other *Chlamydia* proteases, such as Tsp, because previous research has shown that overexpression of Tsp inhibits *Chlamydia* development, and other reports have demonstrated that Tsp is detected only in the inclusion, rather than the cell cytosol [48,49]. We speculate that Tsp targets chlamydial rather than host proteins, and that excessive Tsp may hinder progeny release. Moreover, no studies have suggested that Tsp regulates cell death or participates in *Chlamydia* release during the late stages of infection. Of course, this does not exclude the possibility that Tsp might be involved in this process.

Our study demonstrated a novel mechanism by which *Chlamydia* induces ferroptosis in the late stage of its developmental cycle, highlighting its role in progeny release. Although the *Chlamydia*-infected cell supernatant we detected contained progeny released via both lysis and extrusion approaches, pharmacological inhibition of ferroptosis successfully reduced progeny release, suggesting the important role of ferroptosis in this process.

## Conclusions

This study elucidates a novel mechanism by which *Chlamydia* induces ferroptosis in host cells by degrading SLC7A11 via the chlamydial protease-like activity factor (CPAF). This ferroptotic process plays a crucial role in the release of *Chlamydia* progeny. Notably, pharmacological inhibition of ferroptosis leads in a significant reduction in *Chlamydia* progeny *in vitro*, as well as a decreased burden in the lower genital tract of mice, alongside a trend toward attenuated pathology. These findings provided new insight into *Chlamydia* pathogenesis.

## Supporting information

**S1 Fig.  Temporal dynamics of *Chlamydia trachomatis* growth in HeLa-229 cells.** Immunofluorescence analysis of chlamydial HSP60 in *Chlamydia trachomatis* serotype D-infected cells was performed over a time course.
(TIF)

**S2 Fig.  CPAF-deficient strain exhibits reduced progeny yield.** (A) The cryptic plasmid copy number and chlamydial HSP60 levels in the cell supernatant of CPAF-deficient strain (L2-17/

mCherry) (MOI 1)-infected HeLa-229 cells were measured, compared to those in CPAF-supplemented strain (L2-17/CPAF) (MOI 1)-infected cells. Statistical analysis was performed using a Student's t-test (n=3). (B) The *ompA* copy number in the total culture (supernatant and monolayer) of CPAF-deficient strain (L2-17/mCherry) (MOI 1)- or CPAF-supplemented strain (L2-17/CPAF) (MOI 1)-infected HeLa-229 cells was determined over a time course. Statistical analysis was performed using a two-way ANOVA (n=3). *P* values are indicated as follows: *, $P < 0.05$; **, $P < 0.01$; ***, $P < 0.001$.
(TIF)

**S3 Fig. GPx4 is resistant to degradation by CPAF during *in vitro* incubation.** Cell lysates were incubated with recombinant wild-type CPAF and rCPAF(H105A) for 30 minutes. GPx4 was not degraded by rCPAF. Vimentin, a known CPAF post-lysis substrate, and ERK, a non-CPAF substrate, served as positive and negative controls, respectively.
(TIF)

**S1 Table. Primers used in this study.**
(XLSX)

## Acknowledgments

We thank Prof. Guangming Zhong for providing the L2-17/mCherry and L2-17/CPAF strains, as well as the CPAF antibody (100a). We express our gratitude to Dr. Xiaomian Lin and other colleagues at the Dermatology Hospital of Southern Medical University for their valuable assistance and support throughout this study.

## Author contributions

**Conceptualization:** Wentao Chen, Heping Zheng.

**Data curation:** Wentao Chen, Xin Su, Yuying Pan, Han Zhou, Yidan Gao.

**Formal analysis:** Wentao Chen, Xin Su, Yuying Pan, Han Zhou, Yidan Gao.

**Funding acquisition:** Wentao Chen, Yaohua Xue, Lingli Tang, Heping Zheng.

**Investigation:** Wentao Chen, Xin Su, Yuying Pan, Han Zhou, Yidan Gao, Xuemei Wang, Lijuan Jiang, Yaohua Xue, Lingli Tang.

**Methodology:** Wentao Chen, Xin Su, Yuying Pan, Yaohua Xue, Lingli Tang.

**Project administration:** Yaohua Xue, Lingli Tang, Heping Zheng.

**Supervision:** Bao Zhang, Wei Zhao, Yaohua Xue, Lingli Tang, Heping Zheng.

**Validation:** Lihong Zeng, Qingqing Xu, Xueying Yu, Xiaona Yin.

**Visualization:** Wentao Chen, Xin Su, Yuying Pan, Han Zhou, Yidan Gao.

**Writing – original draft:** Wentao Chen, Xin Su, Yuying Pan, Han Zhou, Yidan Gao, Zhanqin Feng.

**Writing – review & editing:** Wentao Chen, Xin Su, Yuying Pan, Han Zhou, Yidan Gao, Xuemei Wang, Lijuan Jiang, Lihong Zeng, Qingqing Xu, Xueying Yu, Xiaona Yin, Zhanqin Feng, Bao Zhang, Wei Zhao, Yaohua Xue, Lingli Tang, Heping Zheng.

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
