## [Decision Letter · Decision Letter 0]

17 Jul 2024

Dear Dr. Zheng,

Thank you very much for submitting your manuscript "Chlamydial protease-like activating factor targets SLC7A11 for degradation to induce ferroptosis and facilitate dissemination" for consideration at PLOS Pathogens. As with all papers reviewed by the journal, your manuscript was reviewed by members of the editorial board and by several independent reviewers. They have all agreed that it is a very interesting paper that presents novel information on CPAF degradation of the host glutamate-cystine antiporter SLC7A11 (SLC) to drive ferroptosis and promote the release of chlamydial infectious particles (EBs). The paper is well written, and the experimental data are presented logically. In light of the reviews (below this email), we would like to invite the resubmission of a significantly-revised version that takes into account the reviewers' comments. Particularly, the following two major issues must be addressed experimentally:

It is difficult to differentiate if the degradation of SLC is an artifactual processing of SLC in dead cells or during the sample processing by the released CPAF. The authors must be aware that several putative substrates of CPAF have subsequently been shown to be artifacts (https://doi.org/10.1371/journal.ppat.1002842 ; Doi: 10.1016/j.micinf.2013.09.008 ). Thus, efforts must be made to ensure SLC processing occurs inside live cells or cells undergoing ferroptosis. Experimental conditions for excluding CPAF-mediated artificial processing can be found in this paper (doi: 10.1128/JB.02087-12 ), which should be applicable here.There are several concerns about the vitamin E animal data. First, medroxyprogesterone, a long-lasting progesterone, is typically used to synchronize the estrus cycle and sensitize mice to chlamydiae. However, the authors used regular progesterone, which has a half-life of less than an hour in mice. Is this a typo? The detected copy numbers are low for successful C. muridarum (CM) infection. Were the mice infected with CT instead of CM? Regardless, providing recoverable IFU data in vaginal swabs would be more convincing. A hallmark of successful CM infection is the formation of hydrosalpinx. Where are the hydrosalpinx data? The statement “CM-infected mice treated with vitamin E exhibited reduced dilation of the uterine horns (Fig. 4C)” is not supported by the pictures. It is important to demonstrate vitamin E's protective effect using macroscopic data (diameter of uterine horns, frequency of hydrosalpinx) and microscopic data (grades of inflammation and fibrosis). Examples for appropriately presenting chlamydia-induced pathology data can be found in references: https://doi.org/10.1371/journal.pone.0095076  and doi: 10.1128/IAI.01171-15 .The three reviewers have pointed out many imprecise, incorrect, or contradictory/conflicting sentences throughout the manuscript. Please address them.

We cannot make any decision about publication until we have seen the revised manuscript and your response to the reviewers' comments. Your revised manuscript is also likely to be sent to reviewers for further evaluation.

Sincerely,

Guangming Zhong

Academic Editor

PLOS Pathogens

David Skurnik

Section Editor

PLOS Pathogens

Michael Malim

Editor-in-Chief

PLOS Pathogens

orcid.org/0000-0002-7699-2064

Reviewer's Responses to Questions

**Part I - Summary**

Reviewer #1: Chen et al. investigated the mechanism underlying the exit of the obligate intracellular bacterium Chlamydia trachomatis from infected host cells at the end of its developmental cycle. Specifically, the authors tested the hypothesis that Chlamydia trachomatis activates ferroptosis through its secreted protease CPAF. While data from cell culture experiments appear to support this novel mechanism, there are notable weakness with critical data, including CPAF-mediated SLC7A11 in cell culture and the effect of vitamin E in animals. The study can be strengthened by addressing several issues.

Reviewer #2: This manuscript by Chen et.al. entitled “Chlamydial protease-like activating factor targets SLC7A11 for degradation to induce ferroptosis and facilitate dissemination” describes a novel mechanism by which Chlamydia exits cells during the developmental cycle. The in vitro experiments are well designed and executed, for the most part, whereas the in vivo experimentation is somewhat rudimentary. The nomenclature and interpretations are somewhat overstated. Addressing some issues listed below might improve the manuscript.

Reviewer #3: In this interesting paper the authors propose that the chlamydia serine protease CPAF is degrading the host glutamate-cystine antiporter SLC7A11 (SLC) to drive ferroptopsis and promote the release of chlamydial infectious particles (EBs). The paper is clearly written, the experiments that were performed seem solid, and I really enjoyed the clear model presented in 4F. That stated, this paper has several critical limitations that make it difficult to differentiate if their model is correct or if the degradation of SLC that is being observed is an artifactual processing of SLC in dead cells by CPAF released from burst inclusions. Notably, several putative substrates of CPAF have subsequently been shown to be artifacts via the explanation I just proposed, and this manuscript seems to have incompletely considered this data and did not control several experiments sufficiently to differentiate. In particular, I had issues with the absence of one step growth curves in several places. This would have allowed me to assess if the treatments were having the specific effects the authors reported or more likely reflected differences due to growth rates. Second, and related to this, the progeny assays that were performed only looks at the supernatants; again, if chlamydia grew more slowly in the compared conditions just looking at the supernatant could yield misleading results. Finally, and most importantly, no data was shown that proved to me that SLC was being processed inside intact cells. Out of all of my concerns, addressing the last point is absolutely critical to justify the conclusions that were presented.

Specific comments

Line 57. Clarify that this organism is the cause of chlamydia (which is the most common bacterial STI) and lymphogranuloma venereum.

Sentence starting line 59: It is unclear how PID and these other outcomes are key to chlamydial dissemination and propagation. These disease manifestations could be detrimental to chlamydia, unaware of data that support either possibility.

Sentence starting line 65: It is not clear if lysis is the only method progeny are released in vivo. Extrusion is ignored, and, notably, this could be a limitation of interpretation of progeny assays employed later in this study.

Line 85: The idea that CPAF is secreted from intact inclusions is wildly controversial. Many of the cited papers have essentially been retracted. It is clear that CPAF can cleave a wide variety of host cell substrates after lysis of the inclusion. This whole paragraph needs to be heavily caveated with the insights from the Tan group’s work (see PMID: 22876181 for example).

Line 262: To get an idea of the kinetics of chlamydial development using the criteria you describe it is absolutely essential that one-step growth curves be performed. A major concern I have about the entire paper is that authors are looking at what is going on 72 h post infection at high MOIs in cycloheximide treated cells. In my experience, every cell in the flasks is going to be heavily damaged by that point.

Figure 1A: I can tell what is going on from the image, I assume the small specks are PI positive? Does not add much.

Line 270-284: My interpretation is here that you have a very subtle increase in ROS that occurs very late in heavily infected cultures, and that inhibitors of ferroptosis have a complementary but very minor effect around the same time point.

Paragraph starting line 285: The experiments are not able to differentiate if “ inhibition of ferroptosis could effectively attenuate the release of CT progeny” from if the inhibitors caused CT to develop more slowly. This could have been addressed by looking at total genomes and or progeny in the remaining cells as well as supernatant. This issue presents in several subsequent figures and is a significant limitation.

Paragraph starting line 294: The western blotting results could just as easily be explained by the degradation of SLC in lysed cells, by CPAF released from lysed inclusions. Again, this experiment is being done very late during infection, there are lots of burst inclusions and CPAF floating around. This could have been tested by clarifying if there were lower levels of intact SLF in cells containing intact inclusions prior to inclusion lysis. Alternately, the blot could have included antibodies for other plasma membrane proteins that CPAF can cleave in vitro but not in vivo.

Sentence starting line 317: Did lactacystin slow down growth or did it have the stated effect? With all of these experiments, where chlamydia is in development is critical for the comparison; ie if you slow development down there will be differing amounts of free CPAF released from lysed cells. BTW, CT expresses additional S41 family proteases, how do you know if it was Tsp or CPAF?

Paragraph starting line 323: Consider all of my comments above and what would happen if the CPAF null and complement strains develop with slightly different kinetics? One step growth curves and or progeny assays that count organisms in the monolayer and free in the sup simultaneously could be used to differentiate this critical question.

Line 431: As the authors’ own data shows, there are lot of dead and lysed cells in the culture by 72 hpi. For the urea experiment to show anything a time-course would be needed. If secreted CPAF is degrading SLC inside intact chlamydia infected cells, urea would make no difference. I suspect is such a time course were performed the authors would see that the onset of SLC degradation was concomitant with the initiation of progeny production. Separately, IFA would show no evidence of SLC degradation inside cells with intact inclusions.

**Part II – Major Issues: Key Experiments Required for Acceptance**

Reviewer #1: 1. The rationale for analyzing cell death at 48 and 72 hpi is unclear. It is well-established that with cycloheximide, most cells infected with C. muridarum, and to a lesser degree with C. trachomatis serovar D and L2, would have lysed by 48 hpi, while lysis in cycloheximide-free cultures is somewhat delayed. It has been stated that cycloheximide was used for the ferrostatin and liproxstatin-1 experiments. Was cycloheximide also used in other experiments?

2. Figures 1A and other images suggest extensive cell lysis at 48 and 72 hpi. However, upon closer inspection, most cells can still be seen. The quality of data can be improved by expanding the image size (e.g., use 2-full columns width for Fig. 1A), reacquiring images as phase-contrast if they are not, using shorter exposure times under white light for better cell visibility, and presenting a separate set of immunostained images showing inclusions with MOIs of 3 and 10 as supplemental data to document chlamydial growth clearly. Images in Figures 1D and 3A should be improved or replaced for the same reason.

3. Western blotting for SLC7A11 cleavage was done with RIPA buffer, which allows post-lysis cleavage. To confirm physiological cleavage in infected cells, use SDS-PAGE sample buffer (which is also often called SDS-PAGE gel loading buffer) with DTT and immediately heat samples to 100℃ to inactivate CPAF (e.g., Patton et al., 2018).

4. There are several issues with the data from the animal experiments. In the chlamydial mouse genital tract infection model, medroxyprogesterone, a long-lasting progesterone, is typically used to synchronize the estrus cycle and sensitize mice to chlamydiae. However, authors used regular progesterone, which has a half-life of less than an hour in mice. This is unlikely to have a lasting effect. The detected copy numbers seem low, raising the question of successful infection establishment. Providing recoverable IFU data in vaginal swabs would be more convincing. A hallmark of successful C. muridarum infection is the formation of hydrosalpinx. When repeating the study, present hydrosalpinx data. The statement “CM-infected mice treated with vitamin E exhibited reduced dilation of the uterine horns (Fig. 4C)” is not supported by the pictures. To convincingly demonstrate a protective effect of vitamin E, present both macroscopic data (diameter of uterine horns, frequency of hydrosalpinx) and microscopic data (grades of inflammation and fibrosis). Considering the variation in uterine horn thickness shown in Fig. 4C, there may be a need to increase the number of animals per group.

5. P-values are from experiments with multiple groups. Have any statistical procedures been performed to control the false discovery rate?

Reviewer #2: 1. Title: Chlamydia protease-like activity factor is generally the term described in literature. Please clarify the name.

2. Fig 1G: Whereas DMSO is used as positive control for necrosis and chlamydial release from cells, it would be appropriate to include mock as negative control for statistical comparisons.

3. Fig 3: Dissemination is generally used to describe spread of organisms from one tissue/organ to others. A cell culture demonstration of bacterial release does not demonstrate dissemination. Descriptions should reflect data shown.

4. Rationale for the study: The key rationale for this study is stated as the lack of cost-effectiveness of the currently licensed vaccines (Lines 79-82 and 88-91). The JCVI reference (#9) for this does not contain information supporting this statement. It is unclear how reference #12 supports this point of view. Additionally, there is no data described in this manuscript that demonstrated potential cost-effectiveness of the new vaccine being studied. Therefore, the manuscript does not present data to address the purported gap in knowledge that the current study is intended to address.

5. Fig 1E: Uterine horn chlamydial burden was only assessed on day 14. It would be important to show kinetics of bacterial burden in uterine horn. Perhaps the authors can comment whether they consider spread of organisms from lower genital tract to uterine horn as ascension or dissemination (involving spread via blood/lymph)?.

6. Fig 4: whereas the title implicates CPAF in the described effects, no in vivio experimentation was conducted to demonstrate the same. The authors should consider ways to demonstrate how CPAF contributes to ferroptosis and chlamydial spread in vivo.

In summary, this manuscript describes a novel putative mechanism that chlamydia may employ to spread between tissues/organs. Whereas much of the in vitro experiments lend support to this, in vivo experiments do not fully connect the contribution of CPAF to observed effects.

Reviewer #3: 1. Several experiments would benefit from one step growth curves to show that different growth rates do not explain the given phenotypes.

2. IFA or other evidence that SLC is being degraded inside infected cells that contain intact inclusions.

3. 8M urea experiments should be presented as a time course, with one-step growth curves in parallel. The key question is, does degradation of SLC one start before or after host cells lyse.

**Part III – Minor Issues: Editorial and Data Presentation Modifications**

Reviewer #1: 1. Figure legends should be more informative. For example, include the timing of ferrostatin-1 and liproxstatin-1 treatments.

2. Increase the size of labels in most (and perhaps all) figures, particularly Fig. 3, as there is ample space to expand.

3. Line 168: Specify the “green dye.

4. Line 229: Delete “buffer” before “After.”

5. Line 277: Change “identified” to “detected.”

Reviewer #2: (No Response)

Reviewer #3: See overall review.

PLOS authors have the option to publish the peer review history of their article (what does this mean? ). If published, this will include your full peer review and any attached files.

**Do you want your identity to be public for this peer review?** For information about this choice, including consent withdrawal, please see our Privacy Policy .

Reviewer #1: No

Reviewer #2: No

Reviewer #3: No
---

## [Decision Letter · Decision Letter 1]

20 Mar 2025

Dear Dr. Zheng,

We are pleased to inform you that your manuscript 'Chlamydial protease-like activity factor targets SLC7A11 for degradation to induce ferroptosis and facilitate progeny releases' has been provisionally accepted for publication in PLOS Pathogens.

Best regards,

Guangming Zhong

Academic Editor

PLOS Pathogens

David Skurnik

Section Editor

PLOS Pathogens

Sumita Bhaduri-McIntosh

Editor-in-Chief

PLOS Pathogens

orcid.org/0000-0003-2946-9497

Michael Malim

Editor-in-Chief

PLOS Pathogens

orcid.org/0000-0002-7699-2064

Reviewer Comments (if any, and for reference):

Reviewer's Responses to Questions

**Part I - Summary**

Reviewer #1: The authors have responded my criticisms well and quality of the revised manuscript greatly improved.

**Part II – Major Issues: Key Experiments Required for Acceptance**

Reviewer #1: None.

**Part III – Minor Issues: Editorial and Data Presentation Modifications**

Reviewer #1: None.

PLOS authors have the option to publish the peer review history of their article (what does this mean? ). If published, this will include your full peer review and any attached files.

**Do you want your identity to be public for this peer review?** For information about this choice, including consent withdrawal, please see our Privacy Policy .

Reviewer #1: No

---

## [Editor Report · Acceptance letter]

Dear Mr. Zheng,

We are delighted to inform you that your manuscript, "Chlamydial protease-like activity factor targets SLC7A11 for degradation to induce ferroptosis and facilitate progeny releases," has been formally accepted for publication in PLOS Pathogens.

Best regards,

Sumita Bhaduri-McIntosh

Editor-in-Chief

PLOS Pathogens

orcid.org/0000-0003-2946-9497

Michael Malim

Editor-in-Chief

PLOS Pathogens

orcid.org/0000-0002-7699-2064